# Cross-Modal Conceptualization in Bottleneck Models

**Danis Alukaev**[1†]**, Semen Kiselev**[1]**, Ilya Pershin**[1]**,**
**Bulat Ibragimov**[2§]**, Vladimir Ivanov**[1]**, Alexey Kornaev**[1]**, Ivan Titov**[3‡]
[1]Research Center for AI, Innopolis University   [2]DCS, University of Copenhagen
[3]ILCC, University of Edinburgh
[†]d.alukaev@innopolis.university [§]bulat@di.ku.dk
[‡]ititov@inf.ed.ac.uk

## Abstract

Concept Bottleneck Models (CBMs) (Koh et al., 2020) assume that training examples (e.g., x-ray images) are annotated with high-level concepts (e.g., types of abnormalities), and perform classification by first predicting the concepts, followed by predicting the label relying on these concepts. The main difficulty in using CBMs comes from having to choose concepts that are predictive of the label and then having to label training examples with these concepts. In our approach, we adopt a more moderate assumption and instead use text descriptions (e.g., radiology reports), accompanying the images in training, to guide the induction of concepts. Our cross-modal approach treats concepts as discrete latent variables and promotes concepts that (1) are predictive of the label, and (2) can be predicted reliably from both the image and text. Through experiments conducted on datasets ranging from synthetic datasets (e.g., synthetic images with generated descriptions) to realistic medical imaging datasets, we demonstrate that cross-modal learning encourages the induction of interpretable concepts while also facilitating disentanglement. Our results also suggest that this guidance leads to increased robustness by suppressing the reliance on shortcut features.

## 1 Introduction

The limited interpretability of modern deep learning poses a significant barrier, hindering their practical application in many scenarios. In addition to enhancing user trust, interpretations can assist in identifying data and model limitations, as well as facilitating comprehension of the causes behind model errors. Explainable ML has become a major field, with various methods developed over the year to offer different types of explanations, such as input attribution (Sundararajan et al., 2017; Simonyan et al., 2014), free-text rationales (Camburu et al., 2018) or training data attribution (Koh and Liang, 2017). Moreover, methods pursue diverse

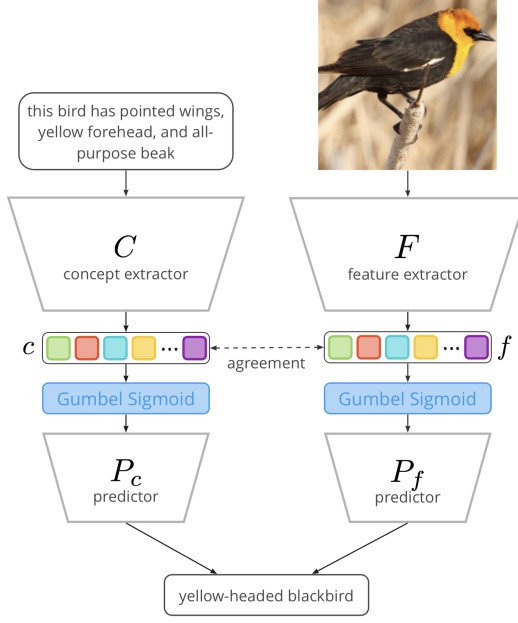

Figure 1: In the XCB framework, during training we promote agreement between the text and visual models' discrete latent representations. Moreover, we introduce sparsity regularizers in the text model to encourage disentangled and human-interpretable latent representations. At inference time, only the visual model is used.

objectives, for example, aiming to generate human-like plausible rationales or extract faithful explanation from the model (Jacovi and Goldberg, 2020; DeYoung et al., 2020).

In this work, our specific focus is on a line of research that aims to develop neural models capable of making predictions via an (semi-) interpretable intermediate layer (Andreas et al., 2016; Bastings et al., 2019; Rocktäschel and Riedel, 2016; Koh et al., 2020). This class of models is appealing because the interpretable component creates a "bottleneck" in the computation, ensuring that all the information regarding the input is conveyed through discrete variables. As a result, these variables play a causal mediation role in the prediction

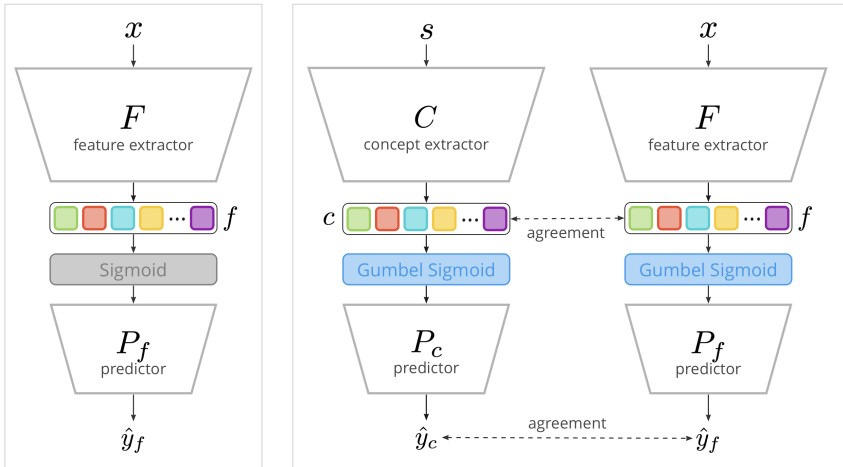

Figure 2: Standard computer vision model w/o interpretability of latent representation $f$ (left), XCB with imposed agreement in visual latent representation $f$ and textual latent representation $c$ (right). Notations in the image: $x$ — visual modality, $f$ — latent representation for $x$, $\hat{y}_f$ — prediction of visual component, $s$ — textual modality, $c$ — latent representation for $s$, and $\hat{y}_c$ — prediction of textual component.

process (Vig et al., 2020). One notable representative of this model class is Concept Bottleneck Models (Koh et al., 2020). CBMs, for a given input image, first predict which high-level concepts are activated, and then infer the class label based on these concepts. This simple and elegant paradigm is advantageous from a practical standpoint as the intermediate layer can be examined, validated and intervened on by an expert user. For instance, in medical diagnostics, the concepts could represent types of abnormalities detected from an X-ray, which makes the model predictions easier to scrutinize and verify for a specialist.

Despite their interpretability advantages, CBMs have several critical limitations. Firstly, they require manual creation and annotation of a set of high-level concepts for each training example, which can be challenging or even unattainable in real-world scenarios. Secondly, concepts designed by domain specialists might not possess sufficient predictive power for classification or could be difficult to reliably detect from the examples (e.g., images).

This work aims to address these shortcomings of CBMs. It considers concepts as discrete binary latent variables, eliminating the need for an expert to predefine the concepts. Rather than relying on the annotation of examples with concepts our approach – Cross-Modal Conceptualization in Bottleneck Models (XCBs)[1] – makes a weaker and arguably more realistic assumption that examples

are linked to texts (such as radiology reports) which serve as guidance for concept induction. Note that the text is not used at test time in our experiments.

Specifically, in training, we use an additional CBM model trained to predict the same target label, but this time using textual data. A regularizer is used to encourage prediction of the same set of concepts in both visual and text CBMs on each training example. In this way, XCB ensures that the concepts can be predicted from both text and images, discouraging the image classifier from relying on features that cannot be expressed through language and, thus, injecting domain knowledge from text into the visual model. Furthermore, an additional regularizer promotes the alignment of concepts with distinct text fragments, thereby enhancing their disentanglement and interpretability.

We conducted experiments involving XCBs on various datasets, including synthetic text-image pairs as well as a more realistic medical image dataset. Our experiments demonstrate that XCB tends to surpass the alternative in terms of disentanglement and alignment with ground-truth (or human-annotated) concepts. Additionally, we incorporated synthetic shortcuts into the images and observed that the inclusion of textual guidance reduced the model's tendency to overly rely on these shortcuts and, thus, made the model more robust.

## 2   Related Work

Concept Bottleneck Models (CBMs) (Koh and Liang, 2017) can be regarded as a representative

---

[1]Released at https://github.com/DanisAlukaev/XCBs

of a broader category of self-explainable architectures (Alvarez Melis and Jaakkola, 2018). In essence, CBMs incorporate a bottleneck composed of manually-defined concepts, which provides a way of understanding 'reasons' behind the model predictions. Examples must be annotated with corresponding concepts, thus, facilitating the training of two distinct segments of the model: the input-to-concepts and concepts-to-label parts.

While we argued that the bottleneck should be discrete to limit the amount of information it can convey, CBMs do not strictly adhere to this principle. In CBMs, even though the concepts themselves are discrete, their representation within the bottleneck layer is not. Specifically, each concept is represented as a logit from a classifier predicting that concept. This deviation from complete discreteness is non-problematic when the two parts of CBMs (input-to-concepts and concepts-to-label) are trained either independently or sequentially. Under such conditions, the logit would not be incentivized to retain input details unrelated to the concepts. Yet, the problem arises when these model parts are trained simultaneously (Margeloiu et al., 2021). Mahinpei et al., 2021 have shown that this information leakage can be alleviated by adopting strictly discrete variables and promoting disentanglement. These insights drove our modeling decisions, prompting us to employ discrete variables and use text (as well as specific forms of regularization) to encourage disentanglement.

A recent approach titled label-free CBM (LCBM) (Oikarinen et al., 2023) focuses on addressing some of the same challenges we address with XCBs, specifically CBM's dependence on predefined concepts and the necessity for manual annotation of examples with such concepts. This study prompts GPT-3 (Radford et al., 2018) to produce a set of concepts appropriate for the domain. Following this, the CLIP model (Radford et al., 2021) is applied to label the images with concepts. In a parallel development, Yang et al. (2023) introduced a similar framework named Language model guided concept Bottleneck model (LaBo), which also incorporates GPT-3 and CLIP as key ingredients. One distinction in their approach is the employment of submodular optimization to generate a set of diverse and discriminative concepts. Although these methods show promise on large visual benchmarks, it is unclear how effectively they would work in specialized domains and how to in-

tegrate additional human expertise in such models. We consider LCBM as an alternative to XCB in our experiments.

Yuksekgonul et al. (2023) introduced a method that posthoc transforms any neural classifier into a Concept-Based Model (CBM). In contrast to their approach, we leverage text to shape the representation inside the neural classifier, potentially making them more disentangled.

In addition to efforts focused on crafting methods for training CBMs, there is an intriguing branch of research dedicated to exploring the utilization of concepts within CBMs to intervene and modify the models (Shin et al., 2022; Steinmann et al., 2023). While this strand of research appears somewhat divergent from our primary interest, it underscores another significant application of CBMs – extending beyond mere prediction interpretation.

## 3 Proposed Framework

In this section we introduce a novel method for cross-modal conceptualization in bottleneck models (XCBs). We start with a high-level overview of the framework and introduce its core components. Further, we consider models used to process visual and textual modalities. Finally, we describe techniques that were used to encourage cross-modal agreement and disentanglement between the concepts.

### 3.1 Overview

The design of the framework (Fig. 1) encourages each element of the discrete latent representation produced by visual encoder to correspond to a single high-level abstraction, which can be expressed by a text span. Accordingly, we use an additional textual model that identifies concepts expressed in natural language and determines the correspondences to visual features.

XCB can tentatively be divided into symmetric visual and textual components (Fig. 2, right). The former consists of a computer vision model $F : x \to f$ (feature extractor) producing latent representation $f$ for an input image $x$ and predictor $P_f : f \to \hat{y}_f$ solving the classification task. Similarly, the textual component includes a natural language processing model $C : s \to c$ (concept extractor) producing latent representation $c$ for a sequence of tokens $s$ and predictor $P_c : c \to \hat{y}_c$. The discreteness of latent variables is imposed by a straight-through version of the Gumbel sigmoid

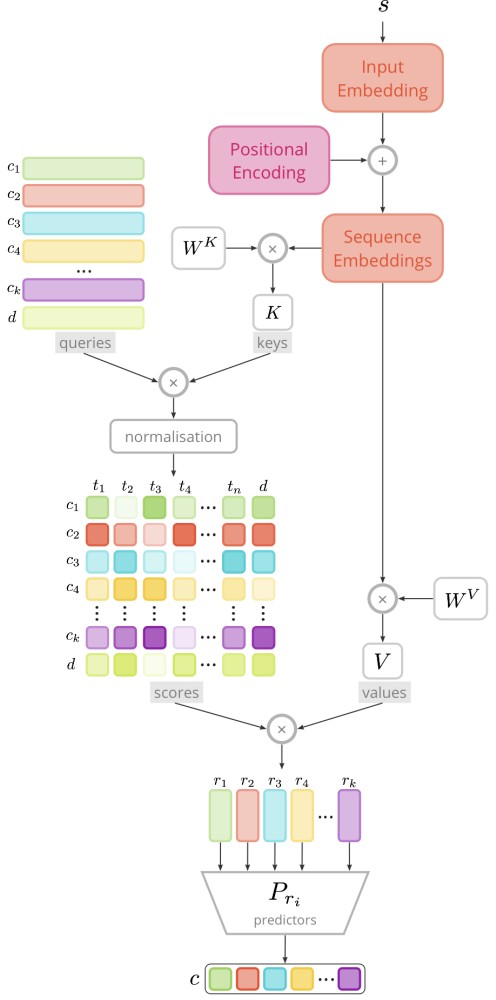

Figure 3: Concept extractor based on cross-attention mechanism. Each query captures semantic of correspondent latent factor in $c$. Tokens with a highest average cross-attention score are regarded as concepts.

(Jang et al., 2017) applied to bottlenecks.

To reinforce agreement in feature and concept extractors at training time latent representations $f$ and $c$ are tied up using a similarity metric. The regularizer encourages model to generate a set of latent factors that could reliably be predicted by both visual and textual encoders.

## 3.2 Unsupervised Approach

The standard computer vision model for classification (Fig. 2, left) is generally composed of the feature extractor $F$ and predictor $P_f$. The latent representation $f$ is an output of a non-linear activation function (e.g., sigmoid). Each factor of $f$ could, in principle, correspond to a single high-level abstraction. Thus, latent representation is not a concept bottleneck per se, but arguably could emulate such behavior. In our experiments we use

such a model with latent sigmoid bottleneck as a baseline and do not impose any pressure to make the representation interpretable or discrete.[2]

## 3.3 Textual Model

The concept extractor is an integral part of XCBs used to process textual descriptions and derive set of concepts. It has been observed that many heads in multi-head attention specialize (Voita et al., 2019), i.e. become receptive to a narrow and unique set of words. Accordingly, we hope that the attention mechanism will reinforce disentanglement in the framework and, thus, use it in the concept extractor $C$. We will discuss below how we can further encourage disentanglement by modifying the attention computation.

Fig. 3 demonstrates the concept extractor adapting cross-attention mechanism for semantic token clustering. Token embeddings are augmented with a sinusoidal positional encoding that provides the notion of ordering to a model (Vaswani et al., 2017). The weight matrices $W^K$ and $W^V$ are used to transform input embeddings into matrices of keys $K$ and values $V$ respectively. We compose a trainable query matrix $Q$ aimed to capture semantics specific to a correspondent factor in latent representation $c$. Averaging matrix of values $V$ with respect to cross-attention scores yields the contextualized embeddings $r_i$, which are further used to estimate elements of $c$ through correspondent predictors $P_{r_i}$.

Since we established one-to-one correspondence between set of queries and latent factors, the matrix of cross-attention scores could be considered as a relevance of each input token to variables in representation $c$. Concept extractor aggregates average cross-attention scores for each token in vocabulary. The concept candidates are, thus, selected by an average attention score for a correspondent latent factor.

The design of concept extractor also accounts for two special cases. First, some token $t$ in the input sequence $s$ might not align with any query (i.e. not correspond to any concept). Second, the opposite, some query might not align with any tokens in the input sequence $s$. We thus introduce additional "dummy" query to capture cross-attention scores from irrelevant tokens, and "dummy" tokens (one

---

[2]In our preliminary experiments, an unsupervised model with a discrete (Gumbel straight-through) layer was not particularly effective, trailing the continuous version according to disentanglement metrics. Thus, we opted to utilize the stronger continuous version as a baseline in our experiments.

per each query) for the model to indicate that all tokens from the input sequence are irrelevant.

## 3.4 Cross-Modal Agreement

To ensure alignment between the visual and textual components in XCB, we propose the inclusion of an additional tying loss function that assesses the similarity between the latent representations, denoted as $f$ and $c$. Since these representations are discrete and stochastic, we employ the Jensen-Shannon divergence to tie the distributions of the visual and textual representations. Given that each representation's distribution can be factored into Bernoulli distributions for individual features, the calculation of the divergence is straightforward.

Because of imposed similarity between representations $f$ and $c$ the set of concepts retrieved by textual component tends to also correspond to visual latent factors, and thereby can be used to explain the visual part. Otherwise, on unseen data, XCB is virtually identical to CBMs: for a given sample $x$ it infers a binary set of human-interpretable attributes $f$, which is further used to predict the target variable $\hat{y}_f$. Note that training with cross-modal agreement yields an additional computational overhead of 15-25% varying by dataset (App. A).

## 3.5 Encouraging Disentanglement

To intensify disentanglement between concepts we borrow ideas from slot attention (Locatello et al., 2020). Instead of normalizing cross-attention scores exclusively along the dimension of tokens, we firstly normalize them along the concepts, and only then rescale scores along the tokens. This method forces concepts (i.e. queries) to compete for tokens, and thus discourages the same token from being associated with multiple concepts. This effect is further amplified by the usage of entmax (Correia et al., 2019) activation function that produces sparse distributions of concepts (for each token).

We also introduce an additional training objective measuring an average pairwise cosine similarity of contextualized embeddings $r_i$ (App. B). This regularization further encourages concepts in the textual component to focus on semantically different parts of an input sequence and, thereby, capture separate factors of variation. Since this procedure is relatively computationally demanding, on each training step the loss is computed based on a subset sampled from $r_i$.

| Parameter | Shapes | CUB-200 | MIMIC |
|---|---|---|---|
| No. of examples | 2,700 | 11,788 | 15,000 |
| No. of attributes | 6 | 312 | 14 |
| No. of labels | 9 | 200 | 2 |
| Vocabulary size | 53 | 8,983 | 8,651 |
| No. of tokens, $\mu$ | 11.2 | 181.2 | 63.0 |
| No. of tokens, SD | 0.8 | 26.4 | 24.8 |
| No. of tokens, max | 12 | 373 | 378 |

Table 1: Description of datasets used in research process. Captions for Shapes and CUB-200 are rule-based generated, whereas in MIMIC captions are structured reports written by radiologists in a free form.

Note that both the promotion of discreteness and disentanglement have been recognized to decrease the inclusion of irrelevant information into concepts, known as concept leakage (Mahinpei et al., 2021). This offers an additional rationale for their application.

## 4 Empirical Evaluation

This section describes the datasets, introduces evaluation metrics for interpretability, and describes the procedure used to measure robustness of models.

### 4.1 Datasets

For our experiments we have used three classification datasets (Tab. 1). Each data point takes a form of $(x, s, a, y)$, where $x$ stands for a visual modality (e.g., chest x-ray image), $s$ — textual modality (e.g., structured medical report), $a$ — binary set of attributes (e.g., types of abnormalities), and $y$ — label of class (e.g., whether a pathology has been detected).

**Shapes.** Primarily for benchmarking we propose a synthetic dataset of primitive shapes (Fig. 4, top).[3] The image $x$ is a white canvas with random shape (i.e., square, triangle, circle) of random color (i.e., red, green, blue), and random size (i.e., small, medium, large). Caption $c$ is generated from visual parameters of a shape, its location, and words sampled from the vocabulary, using a hand-written grammar. The vector of attributes encodes shape and color, their combination represents a target class $y$.

**CUB-200.** Similar to CBMs we evaluate XCB on Caltech-UCSD Birds-200-2011 (CUB-200) (Wah et al., 2011). Due to incoherent intra-class attribute labeling we performed a majority

---

[3]Released at https://github.com/DanisAlukaev/shapes

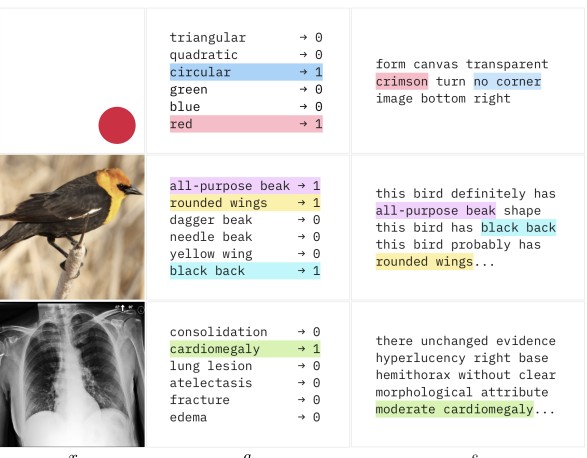

Figure 4: Examples of data points ($x$ — visual modality, $a$ — set of attributes, $s$ — textual modality) from datasets used in research process: Shapes with label "red circle" (top), CUB-200 with label "yellow headed blackbird" (middle), MIMIC-CXR with label "pathology" (bottom).

voting as suggested by Koh et al. (2020). Since CUB-200 does not contain captions, description was generated as provided by the vector of attributes (Fig. 4, middle): each attribute $a_i$ with heading in a form of "has_::<t>" is represented as "this bird has <t> ", e.g., "has_beak_shape::dagger" becomes "this bird has dagger beak shape". The caption for a sample is thereby accumulated from all descriptions of active attributes and tokens sampled from vocabulary.

**MIMIC.** To assess XCB in realistic environment we use a subset from MIMIC-CXR (Johnson et al., 2019). The structured reports are written by radiologists in a free form and without the use of templates. The set of attributes (Fig. 4, bottom) is comprised of characteristics used by radiologists to differentiate pathological cases, which thereby might be considered as underlying latent factors of a certain generative process and used to evaluate latent representations.

## 4.2 Measuring Interpretability

Quantitative evaluation of model interpretability is not entirely standardized. The formulation of a metric highly depends on the specifics of the model architecture and available annotations. Our framework belongs to a set of concept-based models, where structural components are guided by a set of symbolic attributes. In the context of XCBs, we thus define the level of interpretability through

alignment between the latent representation $f$ and vector of ground-truth attributes $a$.

The relationship between latent representation and underlying factors of variation can be quantified in terms of disentanglement, completeness, and informativeness (DCI) (Eastwood and Williams, 2018). Disentanglement indicates degree to which a latent factor captures at most one attribute, completeness reflects degree to which each attribute is captured by a single latent factor, and informativeness accounts for amount of information captured by latent representation. For each attribute $k$, Eastwood and Williams (2018) suggest training linear regressor $R_k : f \to a_k$ with associated feature importance scores $R_k^w$. Disentanglement and completeness are entropic measures derived from $R_k^w$, whereas informativeness is a normalized root mean squared error of $R_k$ computed on the testing subset.

One downside of the DCI metric is the requirement of annotated attributes, which are typically available only for benchmark datasets. Consequently, in a real-world scenario, quality of interpretations could be assessed based on human evaluation. Namely, we provide human experts with a set of images and pieces of text representing underlying concept in the framework. One of the text pieces is sampled from a different concept. The goal of a domain expert is to determine which option is irrelevant. For instance, interpretations of XCB on MIMIC-CXR were evaluated by a practicing radiologist through a set of ten questions. A sample question consists of five X-ray images, with which our framework associated tokens *"pneumothorax"*, *"effusions"*, and *"atelectasis"* (corresponding scaled confidence scores $\psi_i$ were left unknown to the expert). As an additional option, we add *"pneumonia"* and set its confidence score to $\psi_i = 0$. For a selected option $j$ we further define a metric XScore $= 1 - \psi_j$. Higher values of XScore indicate that model interpretation with the least confidence score is more relevant to provided studies than the random one.

## 4.3 Exploring Robustness

Models often learn to rely on non-robust (i.e. spurious) features, which are easy to identify in training but which will not be present or will be misleading at usage time (Ilyas et al., 2019). For example, in the medical domain training data is typically collected from both regular and emergency hospitals. Since most such machines put their serial number

| Modification | $\Delta$ F1-score $\uparrow$ | $\Delta$ Disent. $\uparrow$ | $\Delta$ Compl. $\uparrow$ | $\Delta$ Inform. $\downarrow$ |
|---|---|---|---|---|
| Sigmoid $\rightarrow$ Gumbel Sigmoid | -0.04 ± 0.01 | **0.14 ± 0.02** | 0.06 ± 0.02 | 0.01 ± 0.00 |
| Softmax $\rightarrow$ Entmax | -0.16 ± 0.03 | -0.04 ± 0.00 | 0.05 ± 0.01 | 0.01 ± 0.00 |
| Regular norm. $\rightarrow$ Slot attention norm. | -0.18 ± 0.04 | -0.07 ± 0.01 | -0.03 ± 0.00 | 0.01 ± 0.00 |
| w/o *"dummy"* query & tokens $\rightarrow$ w/ | -0.03 ± 0.00 | 0.03 ± 0.00 | **0.08 ± 0.02** | **-0.02 ± 0.00** |
| Reg. via pairwise similarities of $r_i$ | **-0.01 ± 0.00** | -0.01 ± 0.00 | 0.01 ± 0.00 | 0.00 ± 0.00 |
| $D_{JS}(f'||c') \rightarrow D_{KL}(f'||c')$ | -0.31 ± 0.05 | -0.11 ± 0.02 | -0.03 ± 0.00 | **-0.02 ± 0.00** |
| $D_{JS}(f'||c') \rightarrow D_{KL}(c'||f')$ | -0.27 ± 0.04 | -0.09 ± 0.02 | -0.02 ± 0.00 | **-0.02 ± 0.00** |

Table 2: Ablation study of XCB on Shapes Dataset.

in the corner of the image and emergency hospitals tend to handle harder cases, this ID number becomes a distinctive yet spurious feature (Ribeiro et al., 2016). Reliance of such features is clearly highly undesirable.

To explore the robustness of the XCB model we incorporate a non-robust feature in all training subsets. As a shortcut, we use numerical class label located in the upper left corner of the image. The performance and interpretability of standard and proposed models are further evaluated on samples without distractors. The higher the value of the metric is, the higher robustness is expected.

## 5 Results and Discussion

In this work, we compare the proposed models with the standard 'black-box' model, as well as CBMs and LCBMs (Oikarinen et al., 2023). Due to similarity of LCBM and LaBo (Yang et al., 2023) models, we include in comparison only the former (see Section 2 for their discussion). The experiments were conducted using synthetic and public datasets highlighted in Section 3.1 of the paper. The obtained results are discussed with respect to their performance, interpretability, and robustness.

The ratios between training, validation, and test splits are 75%, 15% and 10%, respectively. For all datasets image size was set to 299 with a batch size of 64. At each epoch, images were normalized and the batches were randomly shuffled. The CUB-200 follows data processing pipeline described in Cui et al. (2018) with additional augmentations such as horizontal flip, random cropping, and color jittering. The hyperparameters were optimized using Tree-structured Parzen Estimator sampling algorithm (Bergstra et al., 2011).

To concentrate on expressivity of concept extractor $C$, we restricted our feature extractor $F$ to be fine-tuned InceptionV3 model (Szegedy et al., 2016) pre-trained on the ImageNet dataset (Deng et al., 2009), with multi-layered perceptron of depth of one for $P_f$ and $P_c$. The baseline model was composed of feature extractor $F$ followed by sigmoid function and predictor $P_f$. The size of latent representations was set to 10, 320, and 20 for Shapes, CUB-200, and MIMIC-CXR datasets respectively (Tab. 1). Textual embeddings were randomly initialized as trainable parameters of length 50. The CBM size of bottleneck matched the number of attributes in a dataset (Tab. 1). To run experiments with CBM[4] and LCBM[5] the official implementations were used. Weights of all models were initialized using Xavier uniform distribution.

The feature extractor $F$ and predictor $P_f$ were optimized jointly using AdaDelta optimizer (Zeiler, 2012) with learning rate 0.25 and rho 0.95. The concept extractor $C$ and predictor $P_c$ were optimized jointly via AdamW optimizer (Loshchilov and Hutter, 2019) with learning rate 0.001, betas of 0.9 and 0.999, and weight decay 0.01. The learning rates in AdamW and AdaDelta were adjusted using OneCycleLR scheduler (Smith and Topin, 2019) with maximal learning rates set to 0.001 and 0.25, respectively. Temperature of the Gumbel sigmoid followed the exponential schedule (Jang et al., 2017). The experiments were conducted using early stopping policy on validation loss with patience of 5 epochs. Five random seeds were used: 42, 0, 17, 9, and 3. Mean and standard deviation for the obtained results were computed over five training runs.

**Ablation study.** Components introduced in XCB that aim to improve interpretability and disentanglement of representation cause drop in performance (Tab. 2). Nevertheless, disentanglement and completeness metrics were substantially improved by discretization of latent representations via Gumbel sigmoid and using "dummy" tokens in slot normalisation respectively. Remarkably, using Kullback-Leibler divergence

---

[4]Available online at https://github.com/yewsiang/ConceptBottleneck

[5]Available online at https://github.com/Trustworthy-ML-Lab/Label-free-CBM

| Dataset | Model | F1 ↑ | Disent. ↑ | Compl. ↑ | Inform. ↓ |
|---|---|---|---|---|---|
| Shapes | Standard | **1.00 ± .00** | .60 ± .05 | .64 ± .10 | **.07 ± .01** |
| | CBM | 1.00 ± .00 | .50 ± .07 | .47 ± .07 | .10 ± .01 |
| | LCBM | .97 ± .01 | .55 ± .04 | .49 ± .02 | .08 ± .01 |
| | XCB | .96 ± .01 | **.78 ± .03** | **.74 ± .12** | **.07 ± .02** |
| CUB-200 | Standard | **.80 ± .01** | .18 ± .02 | .17 ± .02 | .81 ± .01 |
| | CBM | .78 ± .01 | .16 ± .03 | .16 ± .02 | .82 ± .03 |
| | LCBM | .74 ± .02 | .19 ± .02 | .15 ± .01 | .84 ± .02 |
| | XCB | .75 ± .01 | **.21 ± .02** | **.20 ± .02** | **.78 ± .01** |
| MIMIC | Standard | **.77 ± .01** | .03 ± .01 | .01 ± .00 | 1.00 ± .00 |
| | CBM | .75 ± .01 | .02 ± .00 | .01 ± .00 | 1.00 ± .00 |
| | LCBM | .73 ± .00 | .01 ± .00 | .01 ± .00 | 1.00 ± .00 |
| | XCB | .74 ± .00 | **.04 ± .00** | **.03 ± .00** | 1.00 ± .00 |

Table 3: Evaluation results on Shapes, CUB-200 and MIMIC-CXR datasets. XCB model outperforms standard model, CBM, LCBM in terms of interpretability (DCI metrics) maintaining a comparable level of performance.

as a tie loss worsens both performance and DCI, which could indicate that neither modality fully guides induction of implicit high-level abstractions.

**Performance.** The standard model outperforms CBMs on all datasets (Tab. 3), which can be attributed to a trade-off between performance and level of interpretability. Another possible reason is that manually derived set of attributes might not be predictive enough for classification. On the medical dataset LCBM is inferior to other models indicating current limitations in the domain knowledge of large language models. XCB achieves comparable to other variations of CBMs performance.

**Interpretability.** The highest DCI scores (see Section 3.2 of the paper) were achieved with the proposed XCB model that could be attributed to domain knowledge transferred to the model (Tab. 3). High values of both disentanglement and completeness indicate that alignment of latent representation and set of attributes tends towards bijective. Although absolute values of DCI on medical dataset are modest, out survey indicates that XCB possess

a better quality of interpretations in terms of XS-core compared to LCBM: 0.72 vs. 0.66.

The framework associates each factor in latent representation $f$ with a set of textual interpretations (Fig. 5). For example, the images corresponding to the fourth neuron activations cluster in terms of shape, i.e., larger value of logit correspond to squares, whereas smaller represent triangles. Explanations of our model are indeed referring to the shape of a figure: highest relevance is achieved by tokens *"triangular"* and *"quadratic"*.

**Robustness.** Following the procedure presented in Section 3.3 of the paper, the performance and interpretability on non-robust Shapes Dataset were measured. Performance of the standard model on testing subset is lower than for XCB on all datasets (Tab. 4). The DCI of the proposed framework are also superior compared to the standard model. Method of integrated gradients reveals that the baseline mainly focuses on non-robust feature, whereas the proposed framework has higher attribution to the shape of the object (Fig. 6). We speculate that such an increase in robustness can be associated with a textual component of the framework, which regularizes implicit visual abstractions.

Note that while we encourage textual and visual components to align, it is not a hard constraint but rather a soft regularizer, which is used only in

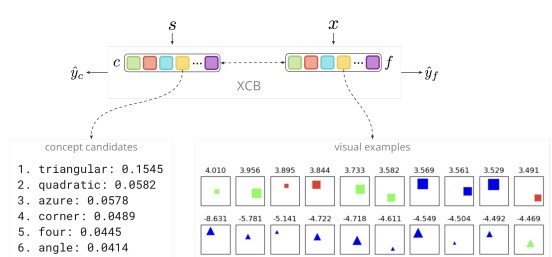

Figure 5: Interpretations for the fourth latent factor on Shapes dataset. Concept candidates for the latent factor (left) describe variance of observed values of logits on bottleneck (right).

| Metric Δ | Shapes | CUB-200 | MIMIC |
|---|---|---|---|
| F1 ↑ | .04 ± .00 | .01 ± .00 | .02 ± .00 |
| Disent. ↑ | .02 ± .00 | .02 ± .00 | .01 ± .00 |
| Compl. ↑ | .02 ± .00 | .02 ± .00 | .02 ± .00 |
| Inform. ↓ | .03 ± .00 | .04 ± .00 | .01 ± .00 |

Table 4: Difference in F1-score and DCI of XCB model compared to standard model on test subset w/o non-robust feature.

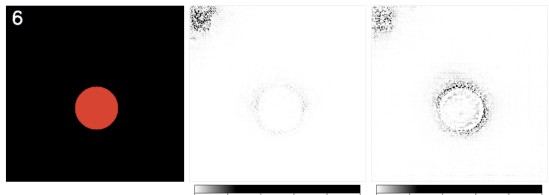

Figure 6: Sample with non-robust feature (left), integrated gradients for standard model (middle), integrated gradients for XCB model (right). XCB shows higher attribution to the shape of an object (robust feature).

| Metric | Shapes | Noisy (10 %) |
|---|---|---|
| F1 ↑ | .96 ± .01 | .94 ± .01 |
| Disent. ↑ | .78 ± .03 | .75 ± .03 |
| Compl. ↑ | .74 ± .12 | .76 ± .12 |
| Inform. ↓ | .07 ± .02 | .09 ± .02 |

Table 5: Performance of XCB on Shapes vs. Shapes with 10% of text descriptions replaced by random ones.

training. Still, systematic errors (e.g., vital concepts consistently omitted in text) could be detrimental. However, a moderate amount of random noise might be less problematic. We tested this hypothesis on the Shapes dataset by taking 10% of the training examples and replacing their text descriptions with random ones. The performance did not change significantly (Tab. 5) confirming our hypothesis.

## Conclusion

The objective of this study is to enhance the practicality of Concept Bottleneck Models by eliminating the need for predefined concepts provided by experts and annotating each training example. Instead, our approach utilizes textual descriptions to guide the induction of interpretable concepts. Through experiments, we demonstrate the advantages of using textual guidance compared in terms of interpretability in a range of set-ups. We believe there are numerous ways to expand on this work, particularly by incorporating a large language model into the textual component. We contend that the broader exploration of leveraging guidance from text or language models to shape abstractions in models for various domains remains largely unexplored. This direction holds great promise for enhancing the transparency and robustness of models utilized in decision-making applications. Our study serves as an illustration of one potential method for providing such guidance.

## Limitations

**Information leakage.** A concept has the potential to reveal additional information about the input, which may not be immediately evident upon inspecting the concept itself. To address this concern, we employ discrete representations and promote disentanglement between concepts, drawing on conclusions from a prior investigation

on information leakage (see section 4 in Mahinpei et al. (2021)).

**Domain dependence.** Intepretation of neurons is known to not generalize across domains (Sajjad et al., 2022): the same neuron may encode different phenomena in two different domains. It is likely to be the case for latent concepts. Our studies were restricted to fairly narrow domains.

**User studies.** The most reliable way to guarantee the interpretability and usefulness of representations is through user studies. However, conducting these studies for evaluating interpretability can be costly and challenging to design. In this work, we primarily focused on automatic metrics instead. Nevertheless, it is important to note that these metrics are not flawless and constrained us to datasets where ground-truth semantic concepts are available.

## Ethics Statement

Like any other machine learning models, the potential for misuse of XCB technology should not be overlooked. Users of XCB, along with other interpretability technologies, need to be aware that it lacks guarantees of faithfulness. Moreover, being a new model architecture, it runs the risk of introducing harmful biases that are not present in more established architectures. Consequently, its application in practical settings necessitates thorough user studies and scrutiny.

Although our experiments employ a public medical dataset, and we hope that this line of research will ultimately yield systems capable of effectively supporting decision-making in healthcare, it is crucial to emphasize that, at the current stage of development, such unproven technology cannot be employed in any critical application.

## Acknowledgements

Danis Alukaev, Semen Kiselev, Ilya Pershin, and Alexey Kornaev were financially supported by The

Analytical Center for the Government of the Russian Federation (Agreement No. 70-2021-00143 dd. 01.11.2021, IGK 000000D730321P5Q0002).

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

## A Computational Requirements

The primary source of computational overhead in XCB (compared to standard models) tends to be the cross-modal agreement of latent representations on training time, which in turn is majorly influenced by the size of bottleneck. Keeping bottlenecks of reasonable size (approximately equal to the number of factors of underlying generative process) results in 15-25% increase in training time (Tab. 6). Note that another possible source of computational complexity could be an average pairwise cosine similarity of contextualised embeddings $r_i$. Although, in practice it was applied to a random subset of 5% of training examples and did not affect the runtime substantially.

## B Optimization

Training objective for XCB models is generally consists of four terms:

- classification loss functions for visual and textual components of XCB, e.g, for all our setups we used regular cross-entropy loss;

- tying loss that ensures cross-modal alignment of two latent representations $c$ and $f$ activated by sigmoid function, e.g., Jensen–Shannon divergence (1);

$$D_{JS}(c'||f') = \left[ m' = \frac{1}{2} \left( c' + f' \right) \right] =$$
$$= \frac{1}{N} \sum_{i=0}^{N} c_i' \cdot \log \left( \frac{c_i'}{m_i'} \right) +$$
$$+ \frac{1}{N} \sum_{i=0}^{N} \left( 1 - c_i' \right) \cdot \log \left( \frac{1 - c_i'}{1 - m_i'} \right) + \quad (1)$$
$$+ \frac{1}{N} \sum_{i=0}^{N} f_i' \cdot \log \left( \frac{f_i'}{m_i'} \right) +$$
$$+ \frac{1}{N} \sum_{i=0}^{N} \left( 1 - f_i' \right) \cdot \log \left( \frac{1 - f_i'}{1 - m_i'} \right)$$

- sparsity regularizer for contextualised embeddings $r_i$, e.g., pairwise cosine similarity (2).

$$loss_{reg}(r) = \frac{1}{n^2} \sum_{i=0}^{n} \sum_{j=0}^{n} \frac{r_i \cdot r_j}{|r_i||r_j|} \quad (2)$$

In practice, visual modality often tends to be more complicated than textual, which makes model favor updating visual component rather than textual one. Empirically, we found that there are two straightforward ways to improve convergence: (a)

| Dataset | Standard | XCB |
|---|---|---|
| Shapes | $30.5 \pm 0.0$ | $35.1 \pm 0.1$ |
| CUB-200 | $85.2 \pm 0.3$ | $106.4 \pm 0.4$ |
| MIMIC | $134.5 \pm 0.4$ | $162.4 \pm 0.5$ |

Table 6: Training time averaged over five training runs for Standard vs. XCB models in seconds on Tesla-V100 16Gb.

pre-train visual component and with a decrease in its learning rate gradually add gradients from textual component, (b) associate each block of the model with a particular frequency of weights updating so that $f_{visual} < f_{textual}$.

## C Examples

**Shapes** (Fig. 7, top): retrieved concepts include description of color ("crimson", "red") and shape ("three", "angles"). Note that concept "circular" corresponds to a negative logit in $f$, so we can speculate that model regards object in the image as the opposite to a "circular" one.

**CUB-200** (Fig. 7, middle): retrieved concepts include description of plumage texture ("buff", "eyering"), beak shape ("hooked", "seabird"), and color of upperparts ("grey").

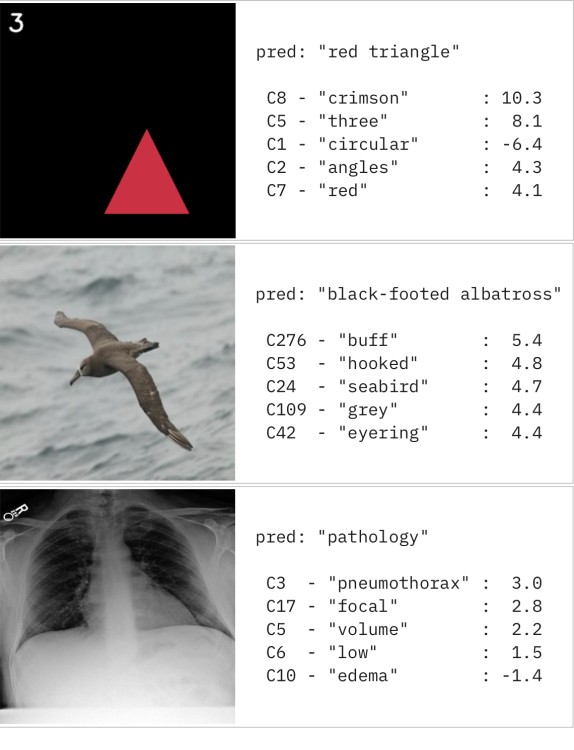

Figure 7: Interpretations of logits in latent representation $f$ on inference time for Shapes dataset (top), CUB-200 (middle), and MIMIC-CXR (bottom) datasets.

**MIMIC-CXR** (Fig. 7, bottom): retrieved concepts include name of observed pathologies ("pneumothorax", "edema"), and some indirect factors ("focal", "volume", "low") used in similar context.

## D  Textual Redundancy

One of the possible applications for XCB models could be medical imaging, where structured reports produced by radiologists often possess redundant data, e.g., in impression and findings sections. To examine how effectively the model could handle and process such data we simulated redundancy by concatenating paraphrased description to the descriptions in the Shapes dataset. We observed no discernible difference in results (Tab. 7) and attribute it to the fact that our sparsity regularizer ensures each text fragment corresponds to a single concept, but does not penalize a concept being inferred from multiple text segments.

| Metric | Original | Redundant |
|---|---|---|
| F1 ↑ | .96 ± .01 | .95 ± .01 |
| Disent. ↑ | .78 ± .03 | .79 ± .05 |
| Compl. ↑ | .74 ± .12 | .80 ± .05 |
| Inform. ↓ | .07 ± .02 | .07 ± .02 |

Table 7: Performance of XCB on original Shapes vs. Shapes with redundant text descriptions.