# OpenReview forum: "Cross-Modal Conceptualization in Bottleneck Models"
_EMNLP/2023/Conference — EMNLP 2023 Main_

### Official Review · Reviewer_RnA7 · 2023-08-05

**Typos Grammar Style And Presentation Improvements:** N/A
**Soundness:** 4

**Excitement:**

4: Strong: This paper deepens the understanding of some phenomenon or lowers the barriers to an existing research direction.

**Missing References:**

N/A

**Paper Topic And Main Contributions:**

an approach for more explainability of the CBM model.
the research paper is about overcoming the limitations of CBM models 1. the need for manual creation and annotation and 2. the fact that concepts designed by domain experts may not be leveraged for prediction tasks. the proposed framework utilizes some guidance from textual data for getting latent representation from the vision encoder. this could encourage the model to get the predicted concepts from both text and images.

**Questions For The Authors:**

1. I'm wondering whether we could see some performance degradation in some scenarios because we are letting the concepts from the vision encoder be learned from both modalities. could this harm the quality of representations? what if the text quality is not good enough or the text is noisy? what if one modality doesn't have good quality?
2. the text may have redundant information, especially in medical datasets. how the model would deal with it?

**Reasons To Accept:**

1. the novelty of matching two modalities' latent representation for CBS models to help have more explainable models.
2. The paper presents a comprehensive evaluation and discussion
3. the paper reads pretty well and clearly states the motivation and proposed idea followed by analysis and discussion

**Reasons To Reject:**

N/A
I have a few questions though


**Reproducibility:**

4: Could mostly reproduce the results, but there may be some variation because of sample variance or minor variations in their interpretation of the protocol or method.

**Reviewer Confidence:**

3: Pretty sure, but there's a chance I missed something. Although I have a good feel for this area in general, I did not carefully check the paper's details, e.g., the math, experimental design, or novelty.

---

> ### Author Rebuttal · Authors · 2023-08-28
>
> Thank you for the positive feedback, and interesting questions.
>
> **Q1: what if the text quality is not good enough or the text is noisy? what if one modality doesn't have good quality?**
>
> Note that while we encourage textual and visual components to align, it's not a hard constraint but rather a soft regularizer, used only in training. Still, systematic errors (e.g., vital concepts consistently omitted in text) could be detrimental (see also our reply to your Q3). However, a moderate amount of `random’ noise might be less problematic. We tested this hypothesis on the Shapes dataset by taking 10% of the training examples and replacing their text descriptions with random ones. The performance did not change that much (XCB on noisy data vs XCB on original data: $0.94$ vs $0.96$ in F1; $0.75$ vs $0.78$ in Disentanglement; $0.76$ vs $0.74$ in Completeness; $0.09$ vs $0.07$ in Informativeness). These are average across 5 runs and we kept the same hyperparameters as the ones used on clean data. We'll describe this experiment in the revised paper. Note also that we experimented with the MIMIC dataset; textual descriptions in MIMIC are written by human experts, so come with ambiguities and inconsistencies.
>
>
>
> That being said, our primary focus is on scenarios where textual descriptions (only available during training) offer guidance to the visual module, see the introduction. The intent is for textual descriptions to instill beneficial biases into the visual component, thereby enhancing its robustness and interpretability. However, if textual data is deceptive – emphasizing irrelevant shortcuts while overlooking crucial concepts – it could be counterproductive. For instance, in Figure 6 and Table 6, the guidance encourages the model to concentrate on the sprite (a presumed robust feature) rather than the background (a presumed distractor). If the background were actually the vital feature and the sprite were a distractor, then that guidance would be misleading.
>
> **Q2: the text may have redundant information, especially in medical datasets. how the model would deal with it?**
>
> We don’t believe redundancy poses challenges for our method. For example, our sparsity regularizer ensures each text fragment corresponds to a single concept but doesn't penalize a concept being inferred from multiple text segments. To empirically verify this, we simulated redundancy by concatenating a 'paraphrased' description to the descriptions in the Shapes dataset and observed no discernible difference in results (XCB on redundant data  vs XCB on original data: $0.95$ vs. $0.96$ in F1; $0.79$ vs $0.78$ in Disentanglement, $0.80$ vs. $0.74$ in Completeness; $0.07$ vs. $0.07$ in Informativeness; all averages across 5 runs). We will add the experiment to the revised paper.

---

### Official Review · Reviewer_K3Jd · 2023-08-05

**Soundness:** 4

**Excitement:**

4: Strong: This paper deepens the understanding of some phenomenon or lowers the barriers to an existing research direction.

**Paper Topic And Main Contributions:**

The paper presents a novel method called Cross-Modal Conceptualization in Bottleneck Models (XCBs) to enhance traditional Concept Bottleneck Models (CBMs). By utilizing text descriptions accompanying images, XCBs automate the concept induction process, avoiding manual annotation. The authors devise a way to synchronize concepts between visual and text data, conducting experiments to show that their approach offers improvements in interpretability, disentanglement of concepts, and model robustness compared to conventional CBMs.

**Questions For The Authors:**

A: How to handle the ambiguities or inconsistencies in the text?

B: What are the computational demands of implementing the proposed method?

**Reasons To Accept:**

The novel XCB for cross-modal learning leveraging both visual and text data to address the significant limitations of conventional conceptual bottleneck models (CBM). By incorporating textual descriptions to guide concept induction, it overcomes the challenges associated with manual annotation and concept definition.

XCB reduces the model's dependence on image shortcuts, making them more robust. This contribution has important implications for building models that are more reliable and less prone to bias or misinterpretation.

This work provides insights into the field of interpretable machine learning. By facilitating the alignment of concepts to different text fragments, the proposed method encourages disentanglement and interpretability of learned concepts.

**Reasons To Reject:**

The success of the proposed method relies on the availability and quality of textual descriptions accompanying images. In cases where such text is incomplete, ambiguous, or of low quality, the performance of the model may suffer.

**Reproducibility:**

4: Could mostly reproduce the results, but there may be some variation because of sample variance or minor variations in their interpretation of the protocol or method.

**Reviewer Confidence:**

4: Quite sure. I tried to check the important points carefully. It's unlikely, though conceivable, that I missed something that should affect my ratings.

---

> ### Author Rebuttal · Authors · 2023-08-28
>
> Thank you for the positive feedback, and interesting questions.
>
> **Q1: How to handle the ambiguities or inconsistencies in the text?**
>
> **Regarding inconsistencies:** Note that while we encourage textual and visual components to align, it's not a hard constraint but rather a soft regularizer, used only in training. Still, systematic errors (e.g., vital concepts consistently omitted in text) could be detrimental. However, a moderate amount of `random’ noise might be less problematic. We tested this hypothesis on the Shapes dataset by taking 10% of the training examples and replacing their text descriptions with random ones. The performance did not change that much (XCB on noisy data vs XCB on original data: $0.94$ vs $0.96$ in F1; $0.75$ vs $0.78$ in Disentanglement; $0.76$ vs $0.74$ in Completeness; $0.09$ vs $0.07$ in Informativeness). These are average across 5 runs and we kept the same hyperparameters as the ones used on clean data. We'll describe this experiment in the revised paper.
>
> **On ambiguity:** Specific types, such as lexical ambiguity, are likely less problematic in specialized domains which motivated this work (e.g., medical records).  Nonetheless, other types (e.g., anaphoric ambiguity) could be challenging for our shallow non-contextualized text encoder. We posit that even if the textual component is wrong on some examples during training, the overall impact may be minimal as long as the majority of cases are straightforward for the text component (similar to random ‘noise’ above).  Still, we agree that it is an interesting question deserving more exploration.  In principle, the concept prediction architecture (Fig 2) can be layered on top of an LLM. This may make textual interpretation of concepts less faithful but could aid in resolving ambiguities.
>
> **Q2: What are the computational demands of implementing the proposed method?**
>
> The overhead is only in training,  since the cross-modal agreement isn't utilized during testing.  Training with cross-modal agreement is 15-25% slower than without it, varying by dataset.  We'll include run times in the revised paper.

---

### Official Review · Reviewer_UqZr · 2023-08-05

**Soundness:** 4

**Excitement:**

3: Ambivalent: It has merits (e.g., it reports state-of-the-art results, the idea is nice), but there are key weaknesses (e.g., it describes incremental work), and it can significantly benefit from another round of revision. However, I won't object to accepting it if my co-reviewers champion it.

**Paper Topic And Main Contributions:**

The paper introduces a novel technique named Cross-Modal Conceptualization in Bottleneck Models (XCBs) to augment the traditional Concept Bottleneck Models (CBMs). By harnessing text descriptions accompanying images, XCBs automate the concept induction process, thus avoiding manual annotation. The authors propose a mechanism to align concepts between visual and textual data. The experiments conducted confirm that their methodology offers advancements in interpretability, disentanglement of concepts, and model robustness over traditional CBMs.

**Questions For The Authors:**

1. How does the proposed model address ambiguities or inconsistencies present in the text?

2. Can the authors shed light on the computational requirements for the execution of the proposed method?

3. In scenarios where one modality is of inferior quality, how does the model ensure that it doesn't negatively affect the quality of representations?

4. Given that textual descriptions, particularly in medical datasets, may possess redundant data, how does the model effectively handle and process such data?

**Reasons To Accept:**

1. The motivation is sound. The approach is well-presented.

2. The experiments and analysis are extensive and show that the proposed approach can boost the Interpretability and robustness of the models.

**Reasons To Reject:**

1. The efficacy of the proposed method is contingent upon the availability and quality of textual descriptions accompanying images. The model's performance might deteriorate if the accompanying text is ambiguous, incomplete, or of substandard quality.

2. There might be scenarios where allowing the vision encoder to learn concepts from both modalities could adversely impact the quality of representations, especially if the quality of one modality is compromised.

3. The textual data, especially in medical contexts, might contain redundant information. It's unclear how the model would effectively manage this redundancy.

**Reproducibility:**

3: Could reproduce the results with some difficulty. The settings of parameters are underspecified or subjectively determined; the training/evaluation data are not widely available.

**Reviewer Confidence:**

3: Pretty sure, but there's a chance I missed something. Although I have a good feel for this area in general, I did not carefully check the paper's details, e.g., the math, experimental design, or novelty.

---

> ### Author Rebuttal · Authors · 2023-08-28
>
> Thanks for the interesting questions and acknowledgement of our detailed experimentation and the clarity of our motivation.
>
>
> **Q1: How does the proposed model address ambiguities or inconsistencies present in the text?**
>
> **Regarding inconsistencies:** Note that while we encourage textual and visual components to align, it's not a hard constraint but rather a soft regularizer, used only in training. Still, systematic errors (e.g., vital concepts consistently omitted in text) could be detrimental (see also our reply to your Q3). However, a moderate amount of `random’ noise might be less problematic. We tested this hypothesis on the Shapes dataset by taking 10% of the training examples and replacing their text descriptions with random ones. The performance did not change that much (XCB on noisy data vs XCB on original data: $0.94$ vs $0.96$ in F1; $0.75$ vs $0.78$ in Disentanglement; $0.76$ vs $0.74$ in Completeness; $0.09$ vs $0.07$ in Informativeness). These are average across 5 runs and we kept the same hyperparameters as the ones used on clean data. We'll describe this experiment in the revised paper.
>
> **On ambiguity:** Specific types, such as lexical ambiguity, are likely less problematic in specialized domains which motivated this work (e.g., medical records).  Nonetheless, other types (e.g., anaphoric ambiguity) could be challenging for our shallow non-contextualized text encoder. We posit that even if the textual component is wrong on some examples during training, the overall impact may be minimal as long as the majority of cases are straightforward for the text component (similar to random ‘noise’ above).  Still, we agree that it is an interesting question deserving more exploration.  In principle, the concept prediction architecture (Fig 2) can be layered on top of an LLM. This may make textual interpretation of concepts less faithful but could aid in resolving ambiguities.
>
> Note that we experimented with the MIMIC dataset; textual descriptions in MIMIC are written by human experts, so come with ambiguities and inconsistencies.
>
>
> **Q2: Can the authors shed light on the computational requirements for the execution of the proposed method?**
>
> The overhead is only in training,  since the cross-modal agreement isn't utilized during testing.  Training with cross-modal agreement is 15-25% slower than without it, varying by dataset.  We'll include run times in the revised paper.
>
> **Q3: In scenarios where one modality is of inferior quality, how does the model ensure that it doesn't negatively affect the quality of representations?**
>
> In this study, our primary focus is on scenarios where textual descriptions (only available during training) offer guidance to the visual module, see the introduction. The intent is for textual descriptions to instill beneficial biases into the visual component, thereby enhancing its robustness and interpretability. However, if textual data is deceptive – emphasizing irrelevant shortcuts while overlooking crucial concepts – it could be counterproductive. For instance, in Figure 6 and Table 6, the guidance encourages the model to concentrate on the sprite (a presumed robust feature) rather than the background (a presumed distractor). If the background were actually the vital feature and the sprite were a distractor, then that guidance would be misleading. As we discussed above (Q1), if the inferior quality refers to being noisy (rather than deceptive / adversarial), we expect this to be less of a problem. There may be ways to detect these issues. E.g., a substantial dip when using cross-modal agreement (i.e., F1 of XCB << Standard) should be a cause for concern. This could indicate shortcuts in the visual data or issues with textual descriptions, warranting a more in-depth investigation.
>
> **Q4: Given that textual descriptions, particularly in medical datasets, may possess redundant data, how does the model effectively handle and process such data?**
>
> We don’t believe redundancy poses challenges for our method. For example, our sparsity regularizer ensures each text fragment corresponds to a single concept but doesn't penalize a concept being inferred from multiple text segments. To empirically verify this, we simulated redundancy by concatenating a 'paraphrased' description to the descriptions in the Shapes dataset and observed no discernible difference in results (XCB on redundant data  vs XCB on original data: $0.95$ vs. $0.96$ in F1; $0.79$ vs $0.78$ in Disentanglement, $0.80$ vs. $0.74$ in Completeness; $0.07$ vs. $0.07$ in Informativeness; all averages across 5 runs). We will add the experiment to the revised paper.

---

### Meta-Review · Area_Chair_oA6t · 2023-09-11

**Recommendation:** 4

**Metareview:**

The paper introduces a novel method known as Cross-Modal Conceptualization in Bottleneck Models (XCBs), aimed at enhancing traditional Concept Bottleneck Models (CBMs). XCBs leverage textual descriptions accompanying images to automate the concept induction process, eliminating the need for manual annotation. The authors present a mechanism for aligning concepts between visual and textual data.

In general, all reviewers agree that this work is well-motivated and sound. The reviewers concur that it holds significant implications for the field of interpretable machine learning. Therefore, the final decision for the paper is acceptance.

---

### Decision · Program_Chairs · 2023-10-07

**Decision:**

Accept-Main

**Comment:**

The paper introduces a novel method known as Cross-Modal Conceptualization in Bottleneck Models (XCBs), aimed at enhancing traditional Concept Bottleneck Models (CBMs). XCBs leverage textual descriptions accompanying images to automate the concept induction process, eliminating the need for manual annotation. The authors present a mechanism for aligning concepts between visual and textual data.

In general, all reviewers agree that this work is well-motivated and sound. The reviewers concur that it holds significant implications for the field of interpretable machine learning. Therefore, the final decision for the paper is acceptance.